# Phenomenological Changes Associated with Deep Brain Stimulation for Obsessive Compulsive Disorder: A Cognitive Appraisal Model of Recovery

**DOI:** 10.3390/brainsci13101444

**Published:** 2023-10-10

**Authors:** Nicola Acevedo, David Castle, Peter Bosanac, Susan Rossell

**Affiliations:** 1Centre for Mental Health, Swinburne University of Technology, Melbourne, VIC 3122, Australia; 2Clinical Services, St Vincent’s Hospital, Melbourne, VIC 3065, Australia; 3Department of Psychiatry, University of Tasmania, Hobart, TAS 7005, Australia; 4Centre for Mental Health Innovation, Hobart, TAS 7005, Australia; 5Statewide Mental Health Service, Hobart, TAS, Australia; 6Department of Psychiatry, University of Melbourne, Melbourne, VIC 3052, Australia

**Keywords:** deep brain stimulation, obsessive compulsive disorder, lived experiences, phenomenological changes, biopsychosocial, self-constructs, dysfunctional beliefs

## Abstract

The current scientific enquiry of deep brain stimulation (DBS) does not capture the breadth of DBS-induced changes to an individual’s life. Considering that DBS is applied in severe and complex cases, it is ethically and clinically necessary to consider the patient perspective and personally relevant outcomes. This lived experience investigation of people with obsessive compulsive disorder (OCD) undergoing DBS aims to provide a comprehensive evaluation of DBS-induced effects associated with OCD psychopathology. Six patients and six carers completed semi-structured open-ended interviews. A blended approach of interpretative phenomenological, inductive, and thematic analysis techniques was employed. Profound psychopathological changes were expressed; individuals felt more alive, had improved cognitive affective control, greater engagement in the world, and were able to manage their OCD. Through suppression of the condition, self-constructs were able to re-emerge and develop. A framework describing the progression of phenomenological changes, and a theoretical model describing changes in the cognitive appraisal of intrusions influencing recovery are proposed. This is the first identified qualitative investigation of DBS-induced changes in psychiatric patients and carers. Findings have implications for patient education and recovery models of OCD, and scientific understanding of DBS effects.

## 1. Introduction

Deep brain stimulation (DBS) is an invasive therapy that modulates various neurocircuitry pathways, and thus, cognitive, affective and/or motor circuits in severe and chronic psychiatric and neurological patients. DBS can be associated with diverse changes in thoughts and behaviours. Yet, the impact of DBS is usually assessed by a discrete clinician-rated scale of a primary symptom domain, which provides a snapshot of clinical changes but does not comprehensively capture the breadth of psychosocial and meaningful impacts [1]. Further, neuroethical concerns around threats to personality and authenticity within DBS applications have been raised [2]. Considering these factors in combination with the complexity of the treatment itself, the patient perspective is critical in the evaluation of DBS efficacy, yet is lacking in the scientific literature. Therefore, we conducted a qualitative investigation on the phenomenological experiences of individuals with obsessive compulsive disorder (OCD) following DBS therapy. First, we introduce the relevant concepts of OCD psychopathology and DBS investigations.

### 1.1. The Cognitive Mechanisms of OCD Psychopathology Are Complex

OCD psychopathology is heterogeneous and reflects multifaceted interacting biopsychosocial mechanisms. Symptoms wax and wane, are affected by environmental triggers or stresses, and can become engrained within an individual’s identity. Patients engage in thoughts and behaviours they do not want to, leading to impaired agency. They mostly have some insight/awareness that their unwanted thoughts and behaviours are irrational, yet still engage in compulsions as the associated tension and anxiety becomes so unbearable (described as akin to being pushed under water). In turn, the emotional state takes precedence over rational thoughts, and individuals strive for unattainable control in their environment. Individuals engage in compulsions with the aim to reduce obsessional anxiety until it ‘feels right’ or a feeling of ‘completion’ is achieved. However, this conditioning of negative reinforcement creates further distress and proliferation of symptoms, in which the desired relief is not achieved [3]. From a cognitive perspective, it is contended that dysfunctional beliefs and maladaptive appraisal of intrusions drive the progression and maintenance of symptoms to a pathological state [4]. Individuals adopt suppression and misappraisal processes to ‘control’ intrusions, yet these habits paradoxically increase the frequency and salience of them.

### 1.2. The Assessment of DBS Efficacy Is Too Narrow

DBS therapy for OCD does not solely target obsessions and compulsions; it leads to common and idiosyncratic changes to one’s thoughts, perceptions, behaviours, beliefs, relationships, and engagement with the world [5,6,7]. Yet, the usual standardised definition of response is operationalised as a 35% improvement on the Yale-Brown Obsessive Compulsive Scale (YBOCS) [8]. Psychiatric patients undergoing DBS are treatment refractory, thus experiencing severe/extreme symptoms across a long disease history. Furthermore, the effects of DBS on complex psychological processes are not immediate, overt, or readily observed objectively by the clinician. Thus, it is necessary to broaden the evaluation of therapeutic response in line with the complexities of the patients’ etiological history and modulatory effects of the therapy. For example by considering individually meaningful goals in recovery and whether these have been achieved [9]. Assessing the effects of DBS by way of a single rating scale does not capture the breadth of profound changes to the person’s sense of self and their way of life. Therefore, it is clinically meaningful and ethically appropriate to consider the broad experience of the patient and psychosocial outcomes following DBS treatment.

### 1.3. DBS Neuroethical Concerns Lack Empirical Evidence

There are several ethical debates within the field of DBS in psychiatry, particularly related to personality, yet these are largely speculative and lack empirical evidence [10]. Personality changes following DBS have been reported across movement disorders and psychiatric conditions [7,11], yet it is necessary to interpret such alterations in keeping with the pathological mechanisms of that patient group. For example, a personality change in an individual with a movement disorder versus in someone with an addiction disorder should be interpreted differently. Whilst often considered an undesirable side effect in movement disorders, in which DBS targets motor networks, such changes may be integral to recovery from OCD, in which DBS targets cognitive-affective-reward networks. Through suppression of pathological thought processes, individuals with OCD gain greater mental clarity to express themselves and behave in a manner that is more akin to their particular desires and needs [7]. Therefore, the main intended outcome of DBS for psychiatric conditions is the modification of unwanted pathological thoughts and behaviours [9], which may indeed be integrated within an individuals’ sense of self. Nevertheless, changes in the person are often viewed as a concern from ethical and clinical perspectives. Yet, use of the term ‘personality’ in this context (and thus, changes to one’s personality) is causing a biased and inaccurate interpretation of changes to the self. The question should not be whether the therapy changes aspects of the person, but at what level such changes occur and how they are perceived by the individual. Further scrutiny from the patient perspective will inform such topics.

### 1.4. The DBS Scientific Literature Lacks the Patient Perspective

Previous qualitative investigations of DBS outcomes in OCD are limited to one centre [6], a case report [5], and indirect evidence from clinicians [10,12]. Furthermore, de Haan and colleagues (2015) collapsed qualitative themes into the four domains within the interview questions, which is not recommended in qualitative analysis. To address this, we employed a more global-inductive and iterative approach, allowing the research question to evolve and a deeper level of interpretation beyond explicit statements.

In previous research, people with OCD have expressed an expanding perimeter of experiences, openness, and responsiveness to their environment, becoming more themselves following DBS, as well as struggling with the burden of normality [5,6,7]. Along the same lines, individuals with major depressive disorder have reported a restoration of pre-morbid personality following DBS [10]. Investigations in people with movement disorders have identified undesirable changes of disinhibition, irritability, impaired decision-making and alienation (reduced authenticity) [10]. Despite these disparities across treatment groups, findings from movement disorder cases are often extrapolated as global, DBS-induced changes, which further add to speculative conclusions. These findings highlight the value in assessing the lived experience of DBS patients, and the need to separate outcomes between movement disorder and psychiatric applications. Moreover, within the context of OCD, family members are often drawn into constant OCD rituals and take on a full-time caring role, therefore providing another critical perspective to understanding changes in symptomology and functioning.

The current study aims to provide a comprehensive evaluation of DBS effects in individuals with OCD, with a view to providing scientific evidence on the spectrum of DBS-induced effects, addressing neuroethical concerns and elucidating the patient perspective.

## 2. Methods

### 2.1. Participants

Individuals participating in our clinical trial investigating DBS for OCD (ACTRN12612001142820) who had reached clinical response were invited to participate in this additional qualitative follow-up study; their closest family members, who had been involved in their care prior to and following DBS therapy, were also invited. Six people with OCD classified as responders to DBS and six close family members participated in the study. Full details of the clinical trial are provided elsewhere. The project was approved by St Vincent’s Hospital Melbourne Human Research Ethics Committee (SVHM HREC 205/21). All participants provided written informed consent.

### 2.2. Procedure

An open-ended and semi-structured interview was conducted one-on-one with the interviewer (N.A.), who had not been involved in the clinical assessments of the original study and was, thus, not known to the participants, avoiding affiliation bias. N.A., a PhD student and clinical trial coordinator, provided an explanatory statement over the telephone with participants to develop rapport and discuss queries relating to the interview process. The interview guides are provided in the Appendix A; all participants were asked the same core questions, while follow-up questions and prompts varied between participants depending on their responses. The interviews were approximately 60 min in duration, and were conducted at Swinburne University, apart from two interviews with carers which were conducted virtually due to travel restraints. Interviews were audio recorded and transcribed verbatim. Transcripts were coded and analysed in the software package, Nvivo 12. Additional details relating to the interview are described in the consolidated criteria for reporting qualitative research (COREQ) items [13] in Appendix A.

The aim of the interview was to explore participants’ lived experience following DBS therapy. The interview questions were developed based on a similar study by [3] that identified changes in OCD DBS patients related to the: (1) self, (2) external world, (3) interpersonal world, and (4) attitudes towards well-being. To adopt a global lens and avoid bias, the interview did not include terms such as personality and symptoms, and, rather, focused on changes to the person and functioning. In the current study, the interview questions and prompts differed from the previous study but were related to the same four themes. Thus, the study investigated the following research question: what are the phenomenological changes experienced by people with OCD, following response to DBS, in relation the self, external world, interpersonal world and attitudes towards well-being? An implicit and latent approach was employed to allow a deeper level of interpretation and minimise distortion or bias. The interview commenced with the individual discussing their life before DBS, and the major changes in their life following DBS, then specific questions relating to the four themes were asked. This provided the interviewer with background on each participant’s life and major themes to re-address later in the interview, while priming the participant to consider their different stages of illness and recovery.

### 2.3. Analysis

Qualitative data were analysed using elements from thematic, content, and interpretative phenomenological analysis (IPA). The approaches share similarities in the iterative and dynamic processes, which require self-reflection and interpretation of the lived experiences of participants. The ways in which the data are interpreted and described vary depending on the approach. Thematic analysis does not commit to a particular theory and is a foundational tool used across a range of methods [14], while IPA is attached to phenomenological epistemology and involves in-depth interpretation and amplification of experiences without distortion or conviction [15], and content analysis involves employing systematic and objective means to make inferences from data [16]. Therefore, advantages of each approach were incorporated to develop a comprehensive qualitative analysis tool to interpret the lived experience of participants in relation to shared phenomena. Within IPA, it is recommended not to collapse the qualitative themes into the themes of the interview (e.g., self, external world, interpersonal world, and attitudes towards well-being). Rather, the themes evolved across the data sets, with limited manipulation. Furthermore, the current method has adopted an inductive (rather than deductive) analytical strategy, allowing the research question to evolve and not committing to a pre-existing framework (i.e., a strict hypothesis or narrow lens). Moreover, latent (rather than semantic) in-depth interpretation was employed, which goes beyond the semantic content and identifies underlying ideas and conceptualisations. The interviewer developed a set of codes and definitions from each transcript in an iterative manner. Participants reviewed the interpretation of the codes, an additional researcher reviewed the codes to ensure consensus (S.R.), and adjustments were made where necessary. A thematic map was then developed that represented the complete set of codes for each transcript (see Appendix A for further details). The thematic map included primary themes, sub-themes and definitions of each. For the purpose of the results and discussion, people with OCD are referred to as ‘participants (P)’ and the family members are referred to as ‘carers (C)’.

## 3. Results

At baseline, the average YBOCS score was 30 ± 3, participants were aged 38 ± 9 years and had a disease duration of 24 ± 9 years. Participants completed the study at different stages of their care and had been receiving DBS therapy between 8 and 99 months (mean 65 ± 31 months). All participants were classified as responders, with a mean YBOCS improvement of 55%. The individual YBOCS changes and other clinical outcomes are reported elsewhere.

Figure 1 shows the thematic map, common themes and sub-themes representative of all individual codes. Themes were classified as psychopathological changes, changes to the self and identity, and changes identified by carers. Each theme and sub-theme are defined and expanded upon in the following section, while extracts are included in the text with examples. Further extracts can be found in Table 1, Table 2 and Table 3. Additionally, psychosocial difficulties experienced through recovery, and changes experienced by carers, are briefly outlined.

### 3.1. Lived Experiences Prior to DBS and Major Changes

Individuals described themselves as previously surrendering, suffocating, suicidal, lost, comatose, stuck, withdrawn, experiencing torture and having no life. P1 felt stuck in their head with constant worry, and experienced continuous suicidal thoughts and intent. This person had a dramatic shift in their mood and mental focus and adopted a more outgoing approach. Furthermore, was engaged in volunteer work, stepped back into their role as a parent, and was motivated to look after their health and enjoy life. P2 was previously reliant on services and family for everyday tasks, generally unable to leave bed and progressively declining. Following DBS, the major changes noticed were greater mental clarity, calmness, and a positive outlook. This person was living independently and engaging in more creative hobbies, although they still required support for everyday tasks. Prior to DBS, P3 had withdrawn from life, was unable to leave the house, stated they had thousands of rituals they engaged in for the duration of their whole waking day, engaged in self-harm to avoid conducting rituals, had undergone 10 in-patient programs, and carried suicidal intent. Now, they viewed themselves as a functioning individual who had overcome everything on their ‘list’ and regained a life. This individual had started a family, which enabled their recovery, although still felt limited in some respects. P4 previously experienced constant panic attacks described as a ‘living hell’, was engaged in compulsions around 70% of the time, was severely depressed and had a previous suicide attempt. This person now demonstrated a great enthusiasm and appreciation for life and was working as a peer support worker and lived experience representative. P5 felt comatose before DBS and believed they had moved past the point of treatment resistance. They described putting in the work during inpatient programs without receiving any relief and stated they had nothing to lose. This person experienced vast changes in their approach to life and engagement in the world, by pursuing outdoor activities, creative hobbies and casual work. P6 was engaged in rituals for around 3 h per day, completing part-time study but unable to work and reliant on family for daily tasks of living, such as cooking. P6 did not identify changes experienced in their life following DBS, apart from a slight reduction in OCD symptoms. It is suspected that a possible primary diagnosis of autism interfered with the individual’s ability to engage in meaningful psychosocial adjustments and articulate these.

Themes below are representative of those who experienced meaningful changes in their day-to-day life. However, there were exceptions to these findings; for example, P6 did not identify meaningful changes, and two carers (C2, C6) expressed limited changes and ultimately had hoped for more, thus representing a sub-group (further addressed in the discussion). Rather than discredit the rich phenomenological changes identified by most of the participants, we have presented the spectrum of changes identified when individuals respond to DBS and are able to adopt meaningful changes in their daily lives.

### 3.2. Phenomenological Themes

#### 3.2.1. Psychopathological Theme 1: More Alive

*1a. Feeling of lightness and being carefree*. Individuals described an immediate feeling of lightness when the DBS was initially turned on, with the exception of P2 who experienced this a few months later when the ‘right’ settings were reached. This uplifting feeling of lightness was associated with a reduction in catastrophising scenarios and a more carefree and calmer state of being.

P5 stated that immediately after the implantation ‘I just felt lighter, there was just this feeling in me, something inside me felt lighter’. P1 stated ‘I feel as if I’m brighter’; it was evident that their calmer state of mind allowed them to feel more open and confident, ‘believe it or not I stand a lot taller than I used to stand…I just felt like I had the whole world on my shoulders’. P1 also expressed the notion of ‘I just don’t care about all that crap’ in relation to the judgement of others, yet would previously feel tormented by this. P3 described their stress response as ‘non-existent’ and perceived this as a ‘good thing’.

Participants expressed some confusion around how to understand this process, yet all described a lighter way of being, which created calmness, openness and additional shifts, suggesting it was an automatic and global change.

*1b. More present.* Individuals were no longer consumed by obsessional thoughts and gained the mental capacity to focus on the present moment. They expressed great appreciation for this newfound ability.

All participants described a reduction in their ‘mental clutter’; for example, P2 stated ‘I couldn’t do this, sit down and have a conversation with someone, I would just be so much in my head, if you asked me a question, I would have to ask you two or three times to repeat the question’. P5 stated they could ‘focus on the real world so to speak’ as they were no longer constantly anxious with obsessional thoughts. Individuals experienced dramatic relief in the ability to focus on things other than OCD. P4 explained this as ‘a completely different ability, to have that capacity to be in the moment, and to have the ability to enjoy life’. Thus, a *reduced focus* on intrusions and obsessions, allowed a *heightened focus* on non-pathological internal and external phenomena. This caused a shift away from a distorted perspective on threat-related information and a better connection with reality.

*1c. Changed outlook and approach to life.* Individuals developed a deterministic attitude to take control of their life, by embracing future opportunities and challenges. From a life previously confined to their bedroom with little, if any concept of the outside world, this shift in perspective represents a dramatic improvement.

Individuals demonstrated the motivation and strength to take control of their situation; for example, P2 stated ‘you think the four walls are your life, but your life is really outside’. P5 also demonstrated this attitude: ‘in hospital there are only 4 walls… then you think- is this my life-nah, and you’ve got to help yourself’ and later emphasised this point by saying ‘you have the option, be outside or be in hospital, the best thing is to be out’.

Participants developed a curiosity and motivation to explore opportunities in life and achieve new things. P5 also stated ‘I am naturally inquisitive so I will give everything a go once’. Demonstrating the ability to explore interests that were previously suppressed by the OCD, P1 joined a boxing class and, even though they thought this was ‘ridiculous’, they enjoyed trying something new and challenging.

*1d. Appreciation for life.* Rather than dreading each day, individuals developed enjoyment and excitement for a new opportunity at life. This was associated with elevated mood and a more vibrant persona.

From previously thinking ‘it’s never going to end’, P1 expressed that ‘it’s just been joyous, to feel as if life can be wonderful and exciting and interesting and doable’. Individuals expressed dramatic changes in their affect; for example, when concluding the interview, P4 stated ‘I am just so grateful, and I just can never tell them how grateful I am, you just can’t put it into words’.

*1e. Life has been saved.* Two participants explicitly stated that their life had been saved, and others implied this was also the case for them.

P2 referred to themselves prior to DBS as there being ‘no one there’, identifying that DBS had been ‘life changing’, and they ‘wouldn’t be here without the DBS’. Notably, the psychosocial changes experiences by this individual were not as marked as for others, and they desired to do more. Nevertheless, they experienced dramatic changes to their mental clarity and ability to manage obsessional intrusions, demonstrating that this relief alone was significant enough to transform their life.

P3 stated ‘my life had been a misery, that was it, I was done, finished, luckily DBS was there, and I had DBS and obviously now I’m functioning…I have my limitations, but at least now I don’t want to die every day, I have a life, it’s challenging, but I’m making it’. P5 stated they had previously surrendered and now felt as though they were ‘fighting’ and doing what they wanted to do. Therefore, individuals lacked reason and motivation to live prior to DBS, whereas now they were living independent and meaningful lives, with some struggles and impairments, but also achievements and happiness.

#### 3.2.2. Psychopathological Theme 2: Improved Cognitive-Affective Control

*2a. Greater mental clarity.* In a comparable manner to the automatic feeling of lightness, all participants described an automatic change in mental clarity, in which the clutter of obsessional thoughts was dramatically reduced.

P2 stated that intrusive thoughts would create a ‘spiders’ cobweb’ in their mind, hindering their ability to pay attention to simple tasks (e.g., music and conversations) as this would cause ‘more noise’. Following DBS, they described ‘it was pretty amazing, like, I had all this stuff beforehand in my head, just all these thoughts, scrambled, and it was just like all the clouds had been lifted’. P3 used to ‘look at a situation in ten thousand different ways and angles’ through an ‘automatic OCD response’, whereas now they were able to be decisive and pay attention to their own thoughts: ‘DBS gave me the ability to think, when I didn’t have DBS I couldn’t think, I couldn’t stop and think’. Thus, through suppression of automatic OCD thoughts, individuals were able to selectively engage their attention; many other changes were rooted within this process.

*2b. Implementation of cognitive control strategies.* Following an automatic shift in mental clarity, individuals were able to better engage in high-order cognitive control processes. Participants could now implement techniques learnt through CBT they had hitherto been incapable of applying, as well as employing exposure therapy to consolidate and expand the benefits from DBS. Individuals also articulated sophisticated insight into these changes, such as describing their triggers, habitual responses, motivating factors and limitations.

P3 clearly articulated that ‘ERP [exposure response and prevention] was impossible to do without the DBS’, which was also the case for other participants. Individuals were also able to separate intrusive thoughts from reality and do ‘the work [CBT]’ they previously found intolerable when the OCD was so prominent.

P2 and P3 described cognitive ‘filtering mechanisms’ to manage intrusions. P2 would shorten all the ‘steps’ associated with a ritual into a simple word that ‘sums up everything for all those steps’; for example, when leaving the house, they could say ‘lockdown and fortress’, whereas before DBS they weren’t able to go outside for an entire year. This individual also described a ‘filing system’ for disturbing images (e.g., blood); putting the thought in a virtual file allowed them to disconnect and move past intrusions. Furthermore, P2 could stop the cascading effect of intrusions by closing their eyes and saying, ‘it’s buried’, creating a ‘fresh slate’ in their mind. P3 described that intrusions would previously come in at 100 km an hour and there was no way to stop them, whereas after DBS they had a mental ‘stop sign’, allowing them to disrupt this process, choose which thought was ‘reasonable’ and further attend to it. The effectiveness of this mechanisms was impacted by their underlying level of stress. Although this process could still cause some distress, it was expressed as a significant alleviation in comparison to being consumed by OCD thoughts.

*2c. Improved self-regulation.* Individuals exhibited improved emotional regulation by stopping automatic and stereotyped reactions to emotionally charged situations and responding more appropriately and calmly.

P2 expressed some residual difficulties related to mood, yet exhibited motivation and insight into emotional regulation processes: ‘my mood is still a bit up and down, but before the DBS it was always down, I always try and look on the bright side, try and stay positive, but my mood would always just be dark’.

This also led to an ability to disconnect from the actual or perceived judgement of others. For example, P1 stated ‘then I realised who cares, if people like you or don’t like you, it doesn’t matter, it doesn’t matter in the real scheme of things’. Furthermore, P4 described themselves as less reactive in difficult scenarios; for example, when managing conflict with their ex-partner, they would think ‘what do I want to get out of this situation… what’s the best way of dealing with that’. Therefore, individuals were able to implement both cognitive and emotional control in a contextually appropriate and flexible manner.

#### 3.2.3. Psychopathological Theme 3: Greater Engagement

*3a. Expanding perimeter of opportunities.* Through an incremental process, individuals developed an awareness of, and ability to engage in opportunities in the world.

P4 stated ‘I feel like the world out there isn’t just for other people anymore, I feel there are possibilities for me now out there too, and that is just amazing’. This person was engaged in several peer support roles, studying, caring for their daughter, and experiencing close and meaningful relationships within their family.

P3 described ‘an expanding horizon opened up’. For two years preceding DBS, they did not leave their bedroom and were merely engaged in computer games, whereas now they were living independently with their partner and child. Despite these changes, and stating they were happy with what they achieved in their recovery, they also expressed some ambivalent views around their engagement with the world. They stated there were more opportunities for them, albeit at the time of the interview they were not seeking them out due to the limitations of their DBS charger.

P5 described simple tasks like going to the supermarket or the pub for a meal as meaningful changes, yet also achieved other complex goals, including socialising and travelling: ‘[I] built up small, so local places and then interstate… so it expands’. Therefore, even relatively minor changes in engagement were meaningful for individuals, and all were able to continually expand their engagement and capabilities in psychosocial functions.

*3b. Able to experience the beauty and joy in the world.* Through shifts in openness and outlook, individuals were able to experience enjoyable elements of life, to which they were previously blunted.

P4 explained ‘it’s beauty that I saw before, and I knew that it was beautiful, but I didn’t feel that it was beautiful… to have the ability to enjoy life, enjoy the little things’. This was described in relation to simply sitting with their daughter, watching the sunset or ocean. This person was going through a process of rediscovering what they enjoyed ‘without the OCD hanging on’. Across participants, this change was related to a greater outward focus, and less so on threat-related information.

*3c. Enriched experiences.* Individuals were not only more engaged with the world, but they also developed deeper and more authentic experiences. Participants exhibited curiosity, enthusiasm, enjoyment and spontaneity in pursuing and engaging in opportunities. They were engaged in artistic, spiritual, and physical activities. This enrichment of experiences also applied to interpersonal connections.

This change was articulated well by P5 who stated, ‘would you prefer to be walking down next to the creek where the birds are chirping or would you rather be in hospital within white walls, you’ve got to help yourself’. Moreover, P2 was now able to pursue interests which were previously too difficult to experience with the constant obsessional thoughts (e.g., ‘noise’), such as meditating, playing basketball, engaging in photography, and going to the movies.

#### 3.2.4. Psychopathological Theme 4: Able to Manage the OCD

*4a. Reduced intensity.* All participants still experienced OCD intrusions and behaviours, yet the intensity was dramatically reduced, allowing them to manage intrusions in their day-to-day lives. Three individuals stated they were still engaged in rituals, yet the amount of time and steps involved were dramatically reduced, and they felt a greater sense of control over OC thoughts and behaviours.

P4 described this process: ‘The OCD symptoms, they are still there, and the compulsions are still there, but it is different, because I don’t get this massive punch… it had the punch taken out of it, the strength taken out of… it was like the monster had been depleted… the intensity had just gone down… I was blown away’.

P5 described the DBS having numbed the OCD, now having 50% of days where they could function well, 25% when the OCD crept in, and 25% ‘really happy days’. The OCD may still ‘ramp up’ or ‘bark back’; for example, if they were tired or stressed, they would be more susceptible to intrusions, or habitual responses, albeit they were no longer overwhelmed with intrusive thoughts or terrified to confront them.

*4b. Took control of OCD.* Through a reduced intensity of OC thoughts, and greater mental clarity, individuals were able to ‘stand up’ and gain control over the OCD.

P5 stated that they ‘defeated’ the OCD, while acknowledging OCD cannot be cured, but it can be effectively managed. P2 found that DBS gave them the ‘power to change things…to say I’m the boss, the OCD is not the boss’, they felt empowered to ‘make up the rules’ and realised that they had the choice, not the OCD.

This process also allowed participants to let go of the unattainable desire to control their environment and place greater trust in others. For example, P3 previously experienced obsessional and irrational fears related to pregnancy, but in their new relationship, they stated ‘with conceiving this child, I had to give up 100% of my control…in terms of my recovery, one of my hardest things on my list of rituals…I accomplished a lot in that sense’.

*4c. New relationship to OCD.* Individuals were able to separate themselves from the OCD, with the exception of P2, who identified more with the OCD, even though they had gained acceptance and distance from their condition.

P5 described they are no longer trying to cure the OCD, but, rather, manage it, and experienced a grieving process of letting go of a frenemy, which they identified with for so long: ‘so you’re grieving for how it used to be…but you have done it for so long, it’s like a loss’. P4 explained the OCD is ‘not nearly as strong anymore…it doesn’t come into a room and fill it anymore. It comes into a room …I go oh you’re here again you bastard, get out…I view it differently…it’s smaller, it’s just smaller.’ P2 accepted the OCD as part of them, although this was a different process to that experienced by others. It allowed them to disconnect and take control. They now identified as stronger than the OCD and capable to ‘make up the rules’.

Individuals also demonstrated improved insight in terms of the irrational nature of intrusions, and related cognitive control processes; for example, P5 stated ‘it’s knowing that it has a mind of its own’.

*4d. Enhanced resilience and strength.* All participants demonstrated strength within themselves and determination to defeat the OCD. They still experienced fluctuations in mood and symptoms, but felt empowered to deal with setbacks, acknowledge bad days as transient, and move forward on their journey to recovery.

P2 stated that before DBS their mood was ‘always dark’, whereas now they could identify that ‘the bad days don’t last’ and that they were ‘a lot stronger since the DBS’. This person could have a good day but still engage in OCD rituals and ‘shrug it off’, demonstrating an ability to overcome the cascading effect of intrusions and distress that would previously occur.

P5 discussed that they would push through bad days and remain committed to social events: ‘I won’t use my OCD as a thing, if I feel crappy or have a compulsion, I will socialise and get through it, without avoiding or excusing myself or walking out, you stick to it’. Therefore, individuals developed the strength to push past difficult situations, which enabled them to progressively take control of their lives.

*4e. On a journey to recovery.* Individuals acknowledged they were on a ‘journey’ to recovery in which they have been empowered to put in the ‘work’. They identified that they will always be working on their recovery, involving ups and downs but an overall progression.

P5 described their journey as climbing a mountain; in which they were previously overwhelmed with the amount of uphill battle that remained, whereas now they were able to look back and appreciate how far they had come: ‘I look at it like a reverse mountain climb, instead of looking at how far you have to go, you look at how far you have come, so you look back and see how far you’ve come and that’s good, and it is a work in progress; that’s how I judge the success of it’. P5 later stated they had ‘got to the top of the mountain’. This person was also focused on acknowledging successes, whether they were small steps or large achievements.

P3 stated they had three recovery points: ‘I got better, and I got better, and I got more better, my life completely changed’ in relation to various milestones in their life they had achieved over years. This individual had been receiving DBS for 9 years and identified that if we had spoken to them earlier, they would have had different answers, thus demonstrating the progressive and continual nature of the DBS therapy and journey of recovery.

Despite dramatic improvements in functioning, individuals still experienced some impairments they wanted to overcome; for example, P4 explained that going out for lunch with friends was enjoyable, but there was still ‘too much anxiety there… I’ve had this disorder that has made me so aware of the terror that you can feel’. Therefore, some of the trauma and distress seemed to be engrained in individuals as somewhat of a ‘hangover’ associated with their previous severity.

#### 3.2.5. Self and Identity Theme 1: Self-Actualization

*Self 1a. Discovery of the self.* Through suppression of OCD thoughts and behaviours, individuals were no longer defined by the disorder and elements of the true self were able to emerge and develop. Individuals described being more outgoing, vibrant, confident, authentic, and having a greater outward focus while understanding themselves better.

P3 explained that they previously identified with individuals who were ‘sick, not functioning, not part of society, not contributing anything, just useless’, whereas now they identified with people who were functioning members of society. Although P4 identified feeling more like themselves, they were still discovering aspects of themselves, for example: ‘I know I love people, but I also get frightened of what people think’.

P1 mentioned that changes to their persona had brought out tension in some relationships; for example, they stated ‘I don’t do things they expect me to do, or I don’t be the way they expect me to be, so they find it a bit, oh that’s a bit confronting’. Thus, participants seemed to be at different points on a trajectory to discovering and understanding themselves more, yet all indicated that they felt more comfortable and more like who they truly were, and none expressed feelings of strangeness.

*Self 1b. Able to appreciate one’s place in the world.* Individuals demonstrated an awareness and appreciation of what they had to offer the world, no longer feeling isolated and debilitated. P4 now valued their experience with mental illness and was working in peer support, which gave them purpose and motivation that they might be ‘able to ease some other people’s suffering’. This person also reflected on her role as a mother and demonstrated pride in terms of how she was raising her daughter: ‘she only had half a mum really… and now I know doing the best I can is actually a really good effort’. P3 felt that they now had something to offer the world, whereas they previously only identified with their ‘sickness’.

*Self 1c. Existential thoughts.* Some expressed concerns in relation to how much time they had left to achieve their goals and desires, representing a vast shift in mindset from previously dreading each day.

For example, P3 was now concerned about the longevity of the device and ‘big picture’ concepts, including whether they would be able to live to an old age to support their family. This individual stated that they previously thought they would be sick for the rest of their life and didn’t think about anything but their ‘sickness’, whereas now they had to consider what ‘normal people’ achieve in their lives. Although this created some internal tension, it demonstrated a wider and prospective view, as well as an appreciation for their role in life.

#### 3.2.6. Self and Identity Theme 2: Prioritization of the Self

*Self 2a. Greater self-worth and self-love.* Individuals developed trust in themselves and were able to appreciate their own worth and capabilities. Also identifying that their condition can manifest as a self-critic and rob one of self-worth, and thus were able to disconnect from this mindset.

Individuals described varying levels of self-criticism, for example P1 stated they didn’t criticize themselves at all, P5 exhibited a strong sense of self-assurance, describing that they found more self-love, and P4 was working on quietening their self-critic, which was still prominent. Although P3 expressed some self-doubt, they were able to acknowledge their achievements: ‘I am proud of myself and I am very happy that I was able to overcome the illness, to a degree’. Similarly, P2 explained ‘I just feel more comfortable within myself, like who I am’. The self-critic for P4 was still ‘very loud’ but they were getting better ‘at fighting it, and quieting it down’, and becoming more comfortable telling themselves they are a ‘good person’.

Overall, participants expressed a new appreciation of themselves and letting go of the unobtainable desire for perfection. Although there was resistance in verbalising this, which was often matched with some critical views.

*Self 2b. Pursuing own wants and needs.* Individuals were no longer surrendering to OCD related demands, and developed a breadth of interests and goals akin to their personal wants and needs.

P5 was now engaging in several artistic activities for therapeutic and enjoyment purposes, such as creative writing, poetry, artwork, hiking, kayaking and travel. This participant stated, ‘I am now doing what I want to do’ and explained ‘I don’t want full-time work because I sort of get bored and I want to move on, I got back into it, I started writing and doing artwork’, and was now selling artwork on a part-time basis. Thus, this person identified what drove them socially and professionally, and pursued opportunities in line with this.

#### 3.2.7. Self and Identity Theme 3: Changed Role in Life

*Self 3a. Improved theory of mind.* Individuals expressed greater empathy and the desire to support others, whereas previously they had disconnected from the experiences and needs of others. Two participants (P4, P5) became passionate about supporting and educating others with severe OCD; one was engaged in lived experience employment, and the other took on a caring role supporting friends with OCD.

Some participants reflected on how their recovery had influenced their close family. P3 expressed ‘I was able to give my dad his life back’, while acknowledging that his recovery would not have been possible without his father. P1 stated they were previously not in ‘tune’ with their daughter’s life, whereas supporting their daughter through mental health struggles of her own was now a primary focus: ‘I want to help her achieve a productive but happy life, as long as it makes her happy. That’s all I want’. Participants were now able to acknowledge the experiences of others close to them and contribute to them in a positive way.

*Self 3b. Gained independence.* Participants no longer required carers and were able to live independent and meaningful lives. The level of psychosocial improvement varied amongst participants. Some had achieved dramatic milestones, such as caring for their child again, starting a family, working as a peer support worker, and travelling, whereas others had a more limited progression. It may be that some are yet to reach these milestones; for example, P3 expressed three major milestones in their recovery across 9 years, stating that ‘all those milestones were hard to reach, but I reached them’.

*Self 3c. Engaged in supporting others.* From previously requiring full-time care and support, individuals were now able to support others and step back into their role as a parent, sibling, daughter, son etc.

P4 found becoming a peer support worker extremely rewarding and developed confidence in the notion that their experience was highly valued by others: ‘the lived experience work I’m doing…I am actually getting paid to talk about what I went through, it’s just incredible’. P2 acknowledged they had improvements to make before they could support their family with health concerns of their own: ‘hopefully I can keep getting healthier and healthier and I can help out more’. P5 stated they were ‘very passionate about educating’ in relation to the lack of understanding around OCD in the general community. Therefore, participants’ empathetic traits were now able to be adopted, as they transitioned into various interpersonal roles.

#### 3.2.8. Carer Theme 1. Changed Outlook

Carer *1a. More present.* Described as the ability to be focused on the present moment and be more carefree, whereas previously participants were consumed with worry and sadness.

C3 described the initial lightness induced by DBS as a ‘light bulb moment, things were different, [they] looked at life differently’. C2 explained that being more present was the major change in the participant’s life: ‘[they are] able to engage in our conversation and not drift away like [they] used to… you could tell when [they] weren’t there, it was like we were walking on eggshells’.

Several carers discussed how improved mental clarity allowed participants to have a more positive and outward focus, whereas prior to DBS they didn’t have the awareness or capacity to focus on anything external as they were constantly in ‘attack mode’ or ‘survival mode’. For example, C5 stated ‘there wasn’t a lot of head space for things other than… survival, rituals, obsessive thoughts’, yet following DBS ‘it was as though a layer of the clouds had lifted’. Therefore, the carers (and participants) identified how internal lightness and improved clarity allowed participants to develop a greater awareness and openness to the external world.

Carer *1b. Acceptance and resilience.* A general acceptance of the condition and other conflicts in life, and the resilience to overcome triggers and setbacks.

Carers identified that participants were now able to manage day to day stressors, such as regular interpersonal conflicts and their remaining OCD intrusions. For example, C5 stated that the participant ‘will still say to me, oh the OCD is bad at the moment, I wouldn’t even have realised it was, so [they] have learnt to manage it incredibly well’. This carer also explained that following the initial ‘lightness’ experienced, the participant then realised they ‘had to challenge the OCD all the time, everyday’, which caused some tension but also acceptance and resilience. Participants were also described as more accepting of others and exercising the ability to let things go and move forward.

#### 3.2.9. Carer Theme 2: Able to Be Their True Self

Carer *2a. No longer defined by OCD.* The true personality and self has been able to emerge and flourish, thoughts and behaviours are no longer dictated by intrusions and obsessions, and participants are able to act in a more authentic manner.

In agreement with the participants, carers described that the OCD was still present, yet participants gained the ability to manage the condition independently with significantly less distress. C5 described that OCD ‘robbed’ the participant of their ability to be social and show their personality, as they were forced to ‘hide away’, whereas DBS ‘allowed [them] to be the person that [they] always have been’. It was identified that changes in mood and energy could influence the participants’ relationship with the OCD, and sometimes it was able to ‘manifest itself more’ (C4). C3 stated that the participant ‘still has remnants of OCD’, yet they were achieving more and becoming happier, and thus becoming more themselves.

Also, carers acknowledged that the OCD no longer infiltrated their lives; the individual still experienced OCD thoughts, yet the intensity and severity was very much diminished. Through suppression of the OCD, the individuals’ true self has been able to shine through and flourish.

Carer *2b. Improved self-regulation and self-worth.* Greater insight into one’s capacity and limits, and the ability to prioritise the self.

An important element of understanding the self involved the awareness that individuals deserved happiness. OCD is associated with substantial guilt, particularly the need for perfectionism and a fear of hurting others; participants were able to move past these beliefs. For example, C4 described that prior to DBS the participant thought of themself as a ‘bad person’, and any improvement would ‘trigger unworthiness’, whereas this was no longer the case.

Several carers identified that participants are acting more in line with their own desires and are no longer driven by the worry of potentially displeasing others. Furthermore, this change was associated with an improved understanding of one’s capabilities (and limitations).

Carer *2c. More vibrant and expressive.* Carers described participants as more vibrant, expressive and energised.

C1 described that they previously only experienced a dulled down or stressed version of the participant, whereas now they were ‘calm, relaxed, decisive, happy, vibrant, much more outgoing, not bogged down in [their] thoughts’, and, therefore, ‘just [themselves]’. C4 described the participant as ‘very fun loving, quite hilarious, very intelligent’, and that these qualities have ‘been given an opportunity to re-emerge’. Thus, true elements of the individuals’ personality were now able to be experienced.

However, C2 described the participant as having a general fatigue associated with being chronically ill; the participant struggled to have the energy to accomplish tasks outside of appointments and sometimes distanced themselves from simple interactions with family.

#### 3.2.10. Carer Theme 3: Enriched Experiences and Relationships

Carer *3a. Experiences joy and happiness.* Carers described the participants as happier, particularly in relation to appreciating the relationships and opportunities they had in their life.

C5 explained that the participant was ‘incredibly grateful’ for the DBS and felt as though they had ‘been given another chance’. C1 described the participant as ‘a lot happier, and in the moment, if something bad happens [they are] like, oh well’.

Carer *3b. More engaged in the world.* An expanding perimeter of engagement in the world, as well as attainment of personally meaningful achievements.

Participants now had the mental clarity to be engaged and interested in what was going in the world. For example, C4 stated ‘it’s been a great joy, to see [their] life enlarging and [them] enjoying things’.

Although all demonstrated greater engagement, the extent varied significantly; C2 discussed the value of support workers visiting the participant to achieve an activity each day (e.g., painting and sport), yet once this was no longer available, the individual declined significantly, and ‘went back into their shell’. Other carers described individuals consistently achieving more in terms of social and occupational activities, including travel, volunteer work, part-time employment, educational courses, social outings, physical exercise, and creative hobbies.

#### 3.2.11. Carer Theme 4: Enabled to Take Control of Life

Carer *4a. Reduced hyperarousal and depression.* Carers described how participants experienced a much calmer and happier state of being and were no longer stuck in a constant loop of hyperarousal and depression.

Although carers did not state that disturbances in mood and anxiety were eradicated, they identified significant improvements, and that participants could independently identify and manage related fluctuations. For example, C4 explained that DBS changed the ‘depth of depression and volcanic effect of anxiety by suppressing the upper and lower limit’; therefore, the level of daily distress was more normalised.

Carer *4b. On a defined path.* The DBS and multifaceted adjunct support empowered individuals to enhance their independence and functioning and utilise their skills and capabilities. In another sense, carers described a continuous upward trajectory throughout recovery, despite occasional setbacks and stressors.

Several carers identified that participants initially tried to achieve too much with their new life, which was maladaptive in the sense that they didn’t have the necessary skills. Carers described that individuals were now better in tune with what goals they could achieve, and the rate at which they could do so. For example, P1 stated that the participant ‘is taking small steps to the bigger picture, whereas before [they] would take giant leaps in different directions’.

C3 described that ‘since the DBS [they have] been able to move on’ in relation to living independently, finding a partner, starting a family, and caring for others. When asked about the overall impression of DBS in relation to the individual’s well-being, it was stated that ‘without the DBS [they] wouldn’t be where [they are] today…[they are] able to have a semi-normal life’.

Thus, despite an initial adjustment period, individuals were described as following a more defined path in achieving their goals.

Carer *4c. Changed role.* Carers described a restoration of an appropriate child-parent relationship; in which support was bidirectional and considered typical in nature.

One participant was able to start a family and care for their partner and child, the carer (C3) stated that having a child was a ‘miracle in itself’ for the individual. This carer and others explained how they were proud of the individual’s ability to step back into their role as a parent, son, daughter, sibling, etc., and had learnt to prioritise others. Furthermore, individuals were described as previously driven by fear or anxiety in relation to pleasing others, whereas now they had adopted more genuine motives, and thus naturalistic engagement styles.

Carer *4d. Personal development with appropriate support.* The ability to mature and achieve developmental milestones, when and if the appropriate support was provided.

This theme is synonymous with the participants’ theme of 4e on ‘a journey to recovery’. However, from the carers’ perspective, it was identified that individuals required personalised support in varying degrees to be equipped with the necessary life skills.

C4 clearly summarised the necessity and importance of support that engaged the participant in daily functional goals by stating ‘the OCD locks you in, in terms of your growth, your capacity to develop what are really life skills’. The participant was now ‘flourishing and doing more now than ever’, as DBS ‘re-enables people’, and they were ‘still seeing benefits of the DBS [7 years later], it’s an accrual of capacity’.

Again, there was an exception to this theme, as C2 described a recent decline in the individual’s mood and functioning and a lack of an upward trajectory. Yet this seemed related to an injury preventing exercise and a lack of psychosocial engagement due to withdrawal of support services. C2 discussed that the participant hoped to achieve more and remains ‘stuck’ and ‘fearful’.

### 3.3. Difficulties Experienced through Recovery

#### 3.3.1. Burden of Normality

The burden of normality phenomenon was impacted by two contributing factors; individuals lacked development skills relative to others their age, and experienced disrupted self-concepts. Some individuals (P3, P4) expressed improvements in their recovery as overwhelming; they fell into a trap of doing too much too quickly and then had to re-adjust their goals and develop appropriate social and occupational skills. This phenomenon was not expressed as distressing to individuals; for example, P1 stated ‘I don’t have the skills to adjust to life in general sometimes…you learn as you go along sort of thing’. Furthermore, after identifying with their illness for several years, individuals had to rediscover and understand elements of the self. Some had developed a strong sense of self, and others were experiencing a transition process of better understanding themselves.

#### 3.3.2. Grief for Lost Time

All participants expressed sadness for the years they had lost due to being debilitated by the OCD. P1 stated ‘the only thing I get sad about, that it didn’t come sooner, that it didn’t happen when I was 20 and not 54’. P4 described grief as well as significant sadness for their past self, even though their ‘miracle happened’, their carer also stated they ‘had to come to terms with a lot of loss, there’s grief about what they haven’t been able to do’. Through detaching and moving past their illness, individuals were faced with the realisation they had spent many years withdrawn from a normal life and thus grieved for lost experiences.

#### 3.3.3. Difficulties of Charging Batteries

One participant expressed great discomfort and annoyance with the need to charge their DBS device, and stated that when they were implanted with a rechargeable battery ‘everything changed’. It was contended that the burden of charging impaired their ability to care for their child, and created a constant reminder of their illness and that they had a device implanted in them. This individual stated ‘I’m a battery that recharges every second day’. This person’s carer stated ‘imagine you are an OCD patient, what is the worst thing you can have? Something that you have to do…they have to sit there with this thing reminding them for 3 h, yes I have DBS, I have a thing in my brain. I felt it was a hindrance to them’. Another carer (C2) expressed a similar view, observing a decline in mood and functioning after recharging the device, which disrupts the participant’s ability to plan and engage in tasks. Thus, charging of the device can become an intrusion in itself; this can also threaten self-concepts, particularly for those with vulnerable and conflicted self-beliefs.

#### 3.3.4. Changes Experiences by Carers

Carers also expressed meaningful changes to their life, including (1) no longer being stuck in the OCD loop, and (2) having a burden lifted (quotes included in Table 3). Carers were no longer stuck in a vicious cycle of engaging in rituals, providing reassurance, or-attempts to minimise psychological distress. Therefore, a significant burden was lifted, allowing the carers time and capacity to focus on themselves. C5 articulated this well: ‘we are not in the OCD loop anymore, [they] manage [their] OCD I know it’s still there, but we are not part of it anymore, [they are] just part of the family, whereas OCD was sitting in on our shoulder all the time’. This allowed an emotional burden to be lifted, in which carers were no longer chronically concerned about the participants. P4 stated ‘it has changed our life dramatically… it’s a joy to see [them] now’. Although C2 was now more hopeful for the participant’s future, they remained in a caring role and had ‘hoped for more’. Moreover, prior to DBS, carers described their relationship to the participants as more distanced (e.g., alienated, in survival mode), whereas now they could spend quality time with the individual (without the OCD infiltrating) and enjoy seeing them progress and experience enjoyment in their life.

## 4. Discussion

Through a phenomenological lived experience investigation in people with OCD classified as responders to DBS therapy, we identified major psychopathological changes. Individuals felt (1) more alive, experienced (2) improved cognitive and affective control and (3) greater engagement in the world, and were (4) able to manage the OCD. Profound changes to the self and identity were also expressed, these being (1) self-actualisation, (2) prioritisation of the self, and (3) changed life roles. Carers reported that participants had (1) a changed outlook, (2) the ability to be their true self, (3) enriched experiences and relationships, and (4) were enabled to take control of life.

The only previous phenomenological investigation of people with OCD undergoing DBS therapy which was identified [6] proposed an enactive-affordance based model, in which reduced anxiety and improved mood gave way to increased trust and reliance on one’s abilities, along with greater openness and outward focus on the world, leading to increased spontaneity. Similarities were present between the current study, including changes in anxiety, mood, confidence, outlook, expressiveness, independence, engagement in the world, openness, spontaneity, flexibility, concentration, relationship to compulsions, and attitudes to individuals’ place in the world. We identified additional multifaceted changes to internal processing systems, (re-)emergence of the self and identity, propose a common progression of changes, and a model to describe alterations in the cognitive appraisal of intrusions

### 4.1. Conceptual Framework: Phenomenological Model of DBS Induced Changes

Across phenomenological themes and sub-themes, the magnitude and sequence of each of the changes varied among participants. For example, all participants were more engaged in the world, but the extent of social and occupational capabilities, and thus overall functioning, varied. In this context, we provide a conceptual framework to describe a prevailing progression of phenomenological experiences and overarching interdependencies (Figure 2).

It is proposed that individuals had an initial and automatic shift in the feeling of lightness, improved mental clarity and drastic reduction in the intensity of the OCD, allowing a calmer and more uplifted state of being. A cascade of cognitive and affective changes was then triggered, predominantly the ability to implement more sophisticated cognitive control and flexibility (e.g., filtering mechanisms of intrusions, and being less reactive and more carefree). Individuals developed a changed perspective on life and became more outwardly focused (e.g., acknowledged they had opportunities and something to offer the world, and had joyous and fulfilling experiences). Ultimately, they experienced greater engagement in the world. These changes occurred in conjunction with an expanding perimeter and enriched quality of experiences; individuals were engaging with and achieving more as time progressed and expressed determination to continue their journey of recovery. Participants described that prior to DBS, they did not have the capacity to consider anything outside of their OCD world, yet were now able to view and evaluate themselves and the world from a completely different standpoint. The progression of these global processes commenced with changes in processing of *internal phenomena* (top half of the triangle) which then triggered changes in processing of *external phenomena* (bottom half of the triangle). In reference to self-concepts, individuals no longer identified with their illness, experienced more appropriate evaluations of the self and world, and thus developed their own desires and needs—termed, a discovery the self.

### 4.2. Proposed Cognitive Model of Changes in the Cognitive Appraisal of Obsessional Intrusions

Additionally, a theoretical model is proposed describing how DBS induced alterations in dysfunctional beliefs allowed for more appropriate appraisal of intrusions and contributed to recovery (Figure 3). This model is described with reference to established cognitive models of OCD that postulate that dysfunctional beliefs related to the self (i.e., feared self, self-ambivalence) and the world (i.e., overestimation of threat, inflated responsibility) and impaired reasoning processes (i.e., inferential confusion) are meta-vulnerabilities to OCD [17,18,19,20,21,22,23]. Individuals are more likely to respond to intrusive thoughts, which endanger or contradict aspects of the self (ego-dystonic) because they believe they are revealing their true self [18,24,25]. Thus, the appraisal of intrusions drives the development and maintenance of obsessions into a pathological state. Prior to DBS, it appeared that dysfunctional belief domains and feared or ambivalent self-concepts contributed to the extreme negative appraisal of intrusions and escalation into obsessions and rituals (represented in the red bottom half of Figure 3). It is contended that extreme negative appraisal of intrusions caused an automatic and bottom-up-predominant processing system, leading to reinforcement and exacerbation of maladaptive OCD responses that became deeply engrained over time.

It is contended that profound phenomenological changes experienced by participants (more alive, greater engagement, improved cognitive and affective control, changes to the self and identity) contributed to improved top-down control of OCD thoughts and behaviours (represented in the blue top half of Figure 3). Dysfunctional beliefs relating to the self and world (bottom red half of Figure 3), which represent cognitive vulnerabilities to OCD, started to break down. In combination with adjunct psychotherapy, DBS recipients learnt to appraise elements of the self and world more appropriately (e.g., there are opportunities out there for me, I am a good person, I am in control). They were able to engage in greater higher order, top-down control of intrusive thoughts by implementing strategies learnt through ERP that were previously too difficult to achieve. Therefore, it is contended that individuals developed more appropriate evaluations of the self and world, which enabled them to develop a strong sense of self, break down maladaptive reappraisal processes, and disrupt automatic and ineffective compulsive behaviours. Individuals were no longer consumed by the OCD and were able manage the remaining intrusions in their day-to-day lives. DBS caused an initial shift in the person, allowing a cascade of flow-on effects; as though an initial domino was knocked over in a sequence of dominoes.

Following on, OCD reflects dysfunctional reasoning and logic, and evaluations of the self and world. From a clinical perspective, the patients within our cohort had an average disease duration of 24 years, had trialled numerous in-patient and out-patient programs, and were heavily medicated, yet their symptoms continued. Considering the chronicity, severity, and complexity of psychopathological profiles within this cohort, it is remarkable that DBS and adjunct psychotherapy was able to target both domains of reasoning and logic and self-concepts, which are thought to drive the exacerbation of intrusions to obsessions and rituals. Thus, consideration of both domains of dysfunctional processing may be necessary for recovery from refractory OCD.

### 4.3. Changes to the Self

The use of the term ‘personality changes’ in the DBS literature is misleading and inaccurate. We propose that the personality is not changed from DBS in people with OCD, rather the self is able to develop, emerge and flourish. Therefore, we define changes to the person as those related to self-concepts.

The self is an agent with an executive function and reflexive consciousness; individuals build up a narrative of themselves by using self-awareness. The self arises, modifies and adapts based on experiences, and exerts control of the environment and regulates inner processes, so therefore depends on elements between the physical body and social environment [26]. The self is an important concept across psychopathology and psychotherapy. Self-ambivalence is a suitable treatment target for OCD [23] and is supported here as a treatment target for therapeutic models in post-operative DBS management.

We avoided use of the term personality within our interviews, and asked the patients if they had experienced changes to who they are as a person. Indeed, psychiatric patients undergoing DBS want to change the way they think, feel, and behave. Within our cohort, participants and carers clearly articulated that people with OCD felt more themselves, more comfortable and more confident within themselves. Furthermore, they were better able to act in line with their own desires and needs and discover their identity without the illness consuming and defining them.

### 4.4. Clinical Relevance of Findings

Our findings inform scientific understanding of DBS-induced recovery from OCD and can be applied to clinical care procedures; specifically related to patient education, evaluation of efficacy, and adjunct psychotherapy. We add to the current scientific understanding of DBS induced effects in OCD patients by demonstrating a breadth of profound and multifaceted changes to psychological, cognitive, affective, and social functioning. The results differentiate between automatic changes from DBS and the progression of more complex higher order changes influenced by combined DBS and psychotherapy.

Currently, OCD patients are informed that there is a 60% likelihood their symptoms will improve by 35% from DBS therapy, but this statement does not encapsulate the multitude of changes they can experience. Rather, patients could be informed that if they respond to DBS they are likely to experience a common progression of changes to their thoughts and behaviours. They may feel an instant lightness, mental clarity, and a reduction in the intensity of the OCD, enabling them to engage in CBT and implement cognitive strategies they previously found intolerable. Through suppression of the illness, they may experience a discovery of the self, and be able to develop their own interests and be empowered to pursue these. This explanation is more representative of personally meaningful changes and may provide better hope and motivation.

Clinical assessment of DBS efficacy should consider multiple domains of functioning. We interviewed six participants who were classified as responders based on the YBOCS criteria of response, yet it became evident that two remained functionally impaired. There have been several arguments that the YBOCS is not an appropriate evaluation of DBS efficacy [1,27,28]. We have shown here that other changes (e.g., lightness, greater engagement, and ability to engage in psychotherapy) are more meaningful to an individual’s well-being and quality of life and may provide greater clinical relevance. For example, short term changes of improved mental clarity, feeling of lightness, and reduced intensity of OCD may suggest appropriate DBS settings and readiness for exposure therapy. Changes in dysfunctional beliefs about the self and world, and greater cognitive flexibility may be indicative of therapeutic response. Currently implemented assessment scales do not capture self-experience, social interactions, existential stance, and openness, which are fundamental to the impact of DBS [3]. As such, the assessment of DBS may benefit from functional measures and self-report scales, although further enquiry is necessary.

Along the same lines, it was clearly elucidated that consistent multidisciplinary support that encourages and enhances social, occupational, and physical functioning is critical to individuals’ recovery, and thus, quality of life. There was strong agreement amongst carers and participants that the psychosocial adjunct care provided (see our clinical guideline for further discussion, [29]) was pivotal in enhancing their confidence, awareness and skill set to transition into normal life. Yet, the scientific literature does not discuss this element of post-operative care for OCD DBS patients. Owing to the chronicity, severity and lack of developmental maturation related to this cohort, extensive support is necessary to encourage, challenge and educate individuals to develop adaptive routines, and minimise burden of normality. Therefore, a comprehensive adjunct treatment approach is necessary within this context and should consider elements within and beyond psychosocial therapy, such as insight into dysfunctional belief domains, peer support, and physical health practices.

There are several limitations within the study. Although we aimed to include the perspectives of all patients within the clinical trial, one was experiencing hypomanic symptoms and one had withdrawn, thus inclusion was deemed inappropriate. The study involved a small sample of twelve participants and should be extended across larger cohorts to potentially validate the frameworks/models proposed. Nevertheless, considering the rarity of OCD DBS patients, and the lack of phenomenological investigations across DBS applications, we provide a substantial level of scientific evidence to elucidate the patient perspective. Owing to the scarcity of lived experience evidence in the field we adopted a broad research question relating to global functional changes. Following our proposed theories on the cognitive appraisal of intrusions, progression of DBS induced changes, and re-emergence of the self, future investigations should adopt a more informed and specific approach exploring these elements.

It is not within the scope of the report to discuss possible placebo effects. Yet, considering the baseline severity and chronicity of these individuals, and longevity of the treatment effect (up to 9 years), we contend there were residual placebo effects. Further, it is proposed that the phenomenological changes experienced were an augmented effect of the DBS, adjunct clinical care, and ongoing implementation of cognitive strategies by the participants.

## 5. Conclusions

This study conducted a comprehensive lived experience investigation of people with OCD following response to DBS. We provide novel frameworks for understanding rich phenomenological experiences of DBS mediated recovery from OCD, with reference to existing cognitive models of self-constructs and dysfunctional beliefs. Moreover, we provide triangulation from a carers perspective and a discussion of difficulties faced through recovery. Findings have implications for patient education, evaluation of DBS efficacy and adjunct psychotherapeutic models. We contend that DBS treatment should incorporate a comprehensive multi-disciplinary adjunct program tailored to the individual and that evaluation of success should consider patient-reported outcomes.

## Figures and Tables

**Figure 1 brainsci-13-01444-f001:**
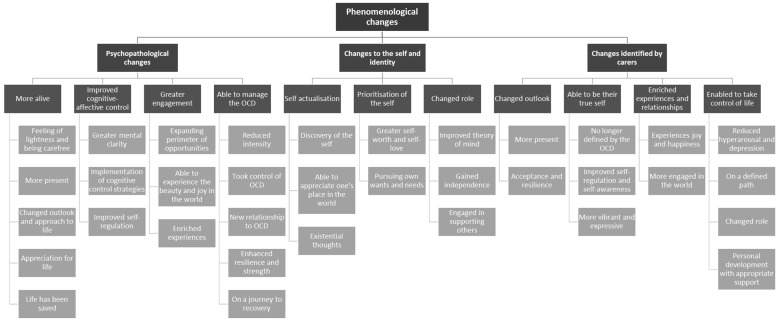
Phenomenological thematic map.

**Figure 2 brainsci-13-01444-f002:**
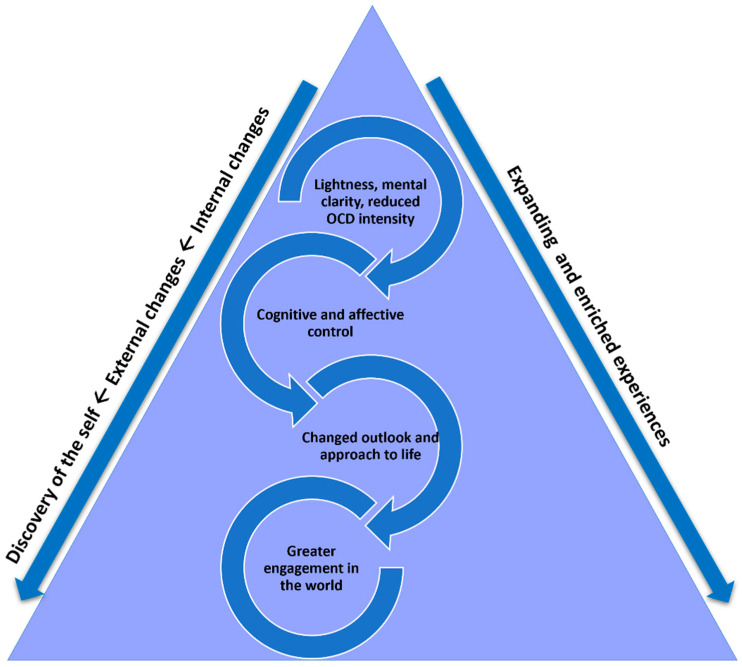
Progression of phenomenological changes expressed by participants.

**Figure 3 brainsci-13-01444-f003:**
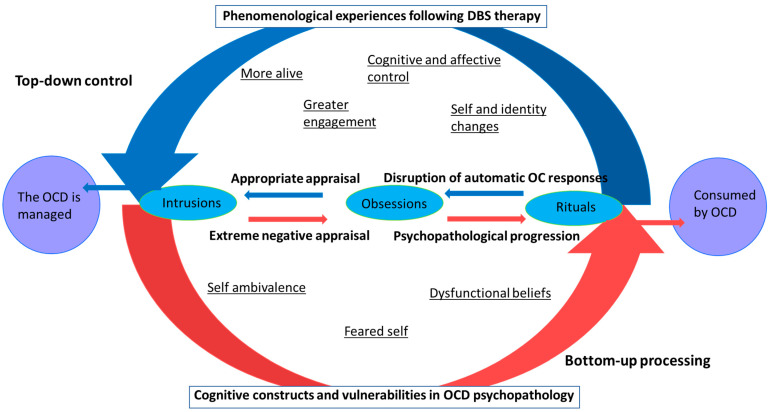
Theoretical framework of the changes in cognitive constructs and vulnerabilities to OCD.

**Table 1 brainsci-13-01444-t001:** Psychopathological phenomenological themes.

Themes	Sub-Themes	Extract
More alive	Feeling of lightness and being carefree	“I just felt lighter, there was just this feeling in me, and something inside me felt lighter” “Immediately I felt better, I just felt better, I don’t know how to explain it” “You could just see in my eyes… there was no one there… but yeah I have more life I guess”
	More present	“A completely different ability, to have that capacity to be in the moment, and to have the ability to enjoy life” “I noticed was the ability to be in the moment…so when I’m just sitting and watching the sunset over the trees or something” “You focus on the real world so to speak”
	Changed outlook and approach to life	“I always try and look on the bright side, try and stay positive, but my mood [before DBS] would always just be dark” “Anything is possible, if I put my mind to it anything is possible, I could try and do anything if I wanted to, nothing is out of my reach” “I started to turn my life around”
	Appreciation for life	“I am just so grateful… you just can’t put it into words” “It’s just been joyous, to feel as if life can be wonderful and exciting and interesting and doable”
	Life has been saved	“I wouldn’t be here without the DBS, that’s for sure” “I felt more alive” “It can change the way you think feel and react and change your life, the DBS has been a lifesaver for me, a life changer” “My life had been a misery, I was done, finished. Luckily DBS was there…now I’m functioning”
Improved cognitive-affective control	Greater mental clarity	“All the clouds had been lifted” “I couldn’t do this, sit down and have a conversation with someone, I would just be so much in my head, if you asked me a question, I would have to ask you 2 or 3 times to repeat the question” “Before DBS I would look at a situation in 10 thousand different ways and angles and stress about it, I have noticed I just let it be” “DBS gave me an ability to think, when I didn’t have DBS I couldn’t think, I couldn’t stop and think” “Now when I have a thought come in that’s stressful, I don’t divert to OCD thinking, that doesn’t happen anymore”
	Implementation of cognitive control strategies	“It’s given me the capacity, to have the confidence, to have the cognitive capabilities, to do the work.” “ERP [exposure response and prevention] was impossible to do without the DBS” “Before the DBS I would have all these steps and now I can sort of, instead of doing every step I can generalise it and just say a word that would sum up everything, for all those steps”
	Improved self-regulation	“I’m better at not reacting… at saying, what do I want to get out of this situation” “There’s been so much that I’ve had to be flexible for, and I’ve just done it, and I’ve smiled all the way through it” “The stress component in my brain doesn’t seem to trigger”
Greater engagement	Expanding perimeter of opportunities	“My worlds gotten bigger… there’s more opportunity” “Before I had no lust for life…. the world has opened up for me, there are opportunities, there are things out there” “I had no opportunities and no way to engage in them before DBS” “I go out and do things that I have never done before” “I feel like the world out there isn’t just for other people anymore…that is just amazing”
	Able to experience the beauty and joy in the world	“I found pleasure in finding lost things that I used to enjoy” “It gets to the point where you think…would you prefer to be walking down next to the creek where the birds are chirping and that or would you rather be in a hospital within white walls. So, you got to help yourself”
	Enriched experiences	“I like to explore and discover things, art, painting, writing” “Social events are really easy- used to be hard, you wouldn’t do it, make up some excuse” “Just being able to have a conversation, hear what they say and take it in… being present, and being part of the friendship or relationship” “I hadn’t hugged my parents for 5–6 years, so I could start to hugging people”
Able to manage OCD	Reduced intensity	“It [OCD] had the punch taken out of it, the strength taken out of….it was like the monster had been depleted.” “It’s not nearly as strong anymore… it doesn’t come into the room and fill it anymore” “Still have about 25% of days where it can creep in, 50% where I can function and 25% where it’s really happy days, whereas before it was virtually 90% shitty days”
	Took control of OCD	“It gave me the power to change things… to say I’m the boss, the OCD is not the boss… I have a choice if I do something 10 times or if I do it once.” “Defeating is a good word, but it [OCD] will bark back sometimes” “I got rid of 99% of my rituals and compulsions, but there was always 1% left…and then I ticked it off”
	New relationship to OCD	“Rituals made me feel safe… definitely a sense of grieving… it was like an enemy, I guess a frenemy, so grieving like a friend of the time” “I’m not trying to cure it anymore, it’s there, I manage it” “You can’t kill OCD, but you can manage it, DBS has definitely helped me manage it”
	Enhanced resilience and strength	“My resilience is a lot better” “When the OCD comes on strong, I am able to sort of shrug it off more easily” “When I am operating at low stress, virtually no [intrusive] thoughts come through and end up as rituals, but when my stress levels are high, the thoughts come through that filter” “I am able to see the bad days don’t last”
	On a journey to recovery	“Still have a way to go, but compared to where I was, much better” “There are good days and bad days, but it’s a work in progress… it’s definitely an upward trajectory” “It’s like a reverse mountain climb, instead of looking at how far you have to go, you look at how far you have come… it’s a work in progress”

**Table 2 brainsci-13-01444-t002:** Self and identity phenomenological themes.

Themes	Sub-Themes	Extract
Self-actualisation	Discovery of the self	“Before DBS I wasn’t able to have a personality, have a sense of self, because my sense of self was my illness, I was my illness. Now I view myself as someone that can do things, not everything, but I can do enough” “I am growing into who I am supposed to be, well not who I am supposed to be, but who I am” “It’s actually a discovery of identity. It’s finding out new things and rediscovering what you enjoy”
	Able to appreciate one’s place in the world	“Now I view myself as, I am a part of society I am a part of the people that do stuff, when your sick you only see sickness and you don’t see ahead” “It [DBS] has given me the ability to start thinking about things other than OCD… to start studying, and a career, and my daughter” “I feel like I’m offering something to the world I couldn’t have done prior to the surgery”
	Existential thoughts	“I have had big picture problems arise where I’ve wanted answers, and I can’t get any answers…what’s going to happen in 30 years” “To feel life is so hard, and when is it going to end, here’s me worrying oh I’ve only got 40 years at the most, where as I used to be- how many more years to go…I wish I had more years to go”
Prioritisation of the self	Greater self-worth and self-love	“I just feel more comfortable within myself, like who I am” “I am getting more comfortable with telling myself, that I’m an okay person, that I am actually a good person” “I am proud of myself, and I am very happy that I was able to overcome the illness, to a degree” “Before DBS I felt terrible and I evaluated myself as not functioning, not functioning, not part of society, not contributing to anything, just useless. Now I feel the opposite”
	Pursuing own wants and needs	“The more I do and the more I achieve and the more I feel connected with people, the better I feel, the better that gets and the quieter the critic gets” “I am now doing what I want to do, this is always what I have wanted to do” “It’s about rediscovering and more focus on me”
Changed role in life	Improved theory of mind	“I feel like I have given my dad his life back… and then started my own” “She only had half a mum… now I know doing the best I can is actually a really good effort…I’m proud of the way I am bringing her up”
	Gained independence	“The thoughts that were there before DBS, I couldn’t have a girlfriend, I couldn’t live alone, I couldn’t have a child… all those milestones were hard to reach but I reached those milestones”
	Engaged in supporting others	“I wasn’t really in tune with her [daughter], now I feel I want to do something, I want to help her achieve a productive but happy life” “The lived experience work I’m doing… is just incredible…people value what you have been through… it makes you feel really good about yourself”

**Table 3 brainsci-13-01444-t003:** Carer phenomenological themes.

Themes	Sub-Themes	Extract
Changed outlook	More present	“It was as though a layer of the cloud had lifted” “It [DBS] enables a person to look at life differently, and able to have a bit more control over your feelings and what you’re doing” “After having DBS I think she is more in the moment, less worried about things that are going to happen in the future, it’s a more kind of normal state of thinking”
	Acceptance and resilience	“I think they did obviously realise that they still had to challenge the OCD all the time, everyday” “Much more accepting of who people are” “If she hadn’t had OCD their life would have been incredibly different, but they have learnt to be incredibly positive about what is in her life now”
Able to be their true self	No longer defined by OCD	“I think it has just allowed her to be the person that she always has been” “She’s definitely more confident out in the world, there’s no question of that, definitely, definitely!” “If the DBS has done anything in terms of her personality it has allowed it to come through” “It’s more like he has an outward focus of 60–70%, but the OCD is still there”
	Improved self -regulation and self-awareness	“She knows sometimes she will feel like not going and still make the effort to go, other times she will say I need to step away from that on this occasion” “It has been a gradual process since the DBS of adjusting to what she can do, and she’s gotten very good at recognising what she can do, but also recognising what she doesn’t want to do”
	More vibrant and expressive	“If she was depressed, she would be sleeping all day or on the coach all day…she’s more awake, she’s more alert, she has more energy” “She’s become more relaxed, more herself and more decisive about things, and yeah, it’s really great to see, she is calm, relaxed, decisive, happy, vibrant, much more outgoing, not bogged down in her thoughts”
Enriched experiences and relationships	Experiences joy and happiness	“She just seems a lot happier, in the moment, if something bad happens she is like- oh well” “She can go off and go out for the evening with her friends and enjoy that, rather than lying in bed and obsessing about different scenarios”
	More engaged in the world	“Well she’s engaged in what is going on in the world, she has a variety of thing, she’s interested in…so she has the headspace now to be interested in that” “I think exercise is a great thing, and he was so motivated, and I loved to hear, when I would ring him, he would tell me, I’ve done this, I am going to rest now and meditate” “She [participant] said …I have been cured from my OCD, I just want to go out and live my life” “It was like a switch, he was doing things, driving around going to places, having mini breaks, stuff like that” “I am happy he has had it done, it has helped him in a lot of ways, he does get out more. Before he was just housebound, before DBS he couldn’t get out of bed, he wasn’t showering”
Enabled to take control of life	Reduced hyperarousal and depression	“Well the DBS has enabled her to keep the OCD under control and to just engage with life in a much more meaningful way” “DBS isn’t an instantaneous cure for these things, but it seems to bring the parameters of the depth of depression and volcanic effect of anxiety, it seems to suppress the upper and lower level” “They felt a reduction in this chronic ambient hyperarousal state”
	On a defined path	“I wouldn’t say by any means she is free of it [OCD]… she has learnt to manage it incredible well” “There have been set back but on the whole it has been a very positive journey, there is no question about it” “What it does, is it re-enables people” “The DBS has allowed her to incrementally take much more control of her life” “She is flourishing and doing more now than she has ever done, and she is working things out” “I’m more hopeful for his future, and I think we can get there, slow and steady but we can get there”
	Changed role	“I think we have a very good relationship, because she has embraced being much more independent” “It kind of made me feel like I was living with a stranger, which I don’t know how to… she changed for the better and I got to see more of the good side of her, which were far and in between but now it is more constant” “I think I’m a bit closer to him then I was before, I felt a bit alienated, it was more, I felt like I was, not a slave, but a go getter boy, or a fix it, whereas now, it’s not as much, he tries to do things himself”
	Personal development with appropriate support	“I think she has matured quite a lot, she has without a doubt, there is no doubt about that, when you are constantly in a state of high anxiety, that undermines any capacity you have for maturing from a child to a teenager to an adult” “Since the DBS he has been able to move on, he left home. he has a partner, and they have a baby, to even have a baby, is a bit of a miracle” “I think that without the DBS I wouldn’t be sitting here talking to you about his family and his achievements that he has made in that period of time” “I can see small changes but I was hoping for a lot more, you know, bigger changes in him”
Themes related to carers	Burden is lifted	“I am much more happier and content now that she’s much more stable” “So it has been a life-saving, a major major event, there is no doubt about it” “It has been a great joy, to see her life enlarging, and her enjoying things, and the relationship she has been able to build with [daughter], which is terrific” “The burden on me is less, less strain, saying you know, does he have enough to eat, does he have his medication, so he takes care of that himself…it has become a blessing, in a way”
	No longer stuck in OCD loop	“We are not in the OCD loop anymore, she manages her OCD I know it’s still there, but we are not part of it anymore, she’s just part of the family, whereas OCD was sitting in on our shoulder all the time” “It has given our lives back to a great extent too” “It’s different now I don’t see the attacks like I used to”

## Data Availability

De-identified data is available upon request.

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
