# Peer review of "Phenomenological Changes Associated with Deep Brain Stimulation for Obsessive Compulsive Disorder: A Cognitive Appraisal Model of Recovery"

_brainsci, 2023, doi:10.3390/brainsci13101444_

Round 1

Reviewer 1 Report

The manuscript titled " Phenomenological changes associated with deep brain stimulation for obsessive-compulsive disorder: a cognitive appraisal model of recovery" presents a comprehensive investigation into the psychopathological changes experienced by individuals with OCD classified as responders to DBS therapy. The study identifies several significant shifts in cognitive, affective, and self-related domains, shedding light on the transformative effects of DBS on their lived experiences. While the manuscript effectively presents its findings, there are areas that require further clarity, refinement, and grammar correction.

1.     Specify the interviewees' relationship to the interviewer in more detail (e.g., researcher's initials, job title).

2.     The manuscript's attempt to provide a conceptual framework and cognitive model is commendable. However, there are instances where sentences are complex and may benefit from simplification. For instance, consider rephrasing "individuals developed a changed perspective and approach to life and thus became more outwardly focused" to "individuals gained a new perspective on life and became more outwardly focused."

Overall, the text is well-structured and presents important concepts related to DBS and its impacts on patients with OCD. The grammar and writing style are generally good, but some sentences could be made clearer by breaking them down into shorter sentences. 

Minor editing of the English language required.

Author Response

Reviewer 1

The manuscript titled " Phenomenological changes associated with deep brain stimulation for obsessive-compulsive disorder: a cognitive appraisal model of recovery" presents a comprehensive investigation into the psychopathological changes experienced by individuals with OCD classified as responders to DBS therapy. The study identifies several significant shifts in cognitive, affective, and self-related domains, shedding light on the transformative effects of DBS on their lived experiences. While the manuscript effectively presents its findings, there are areas that require further clarity, refinement, and grammar correction.

Thank you for the valuable recommendations, the suggestions have been implemented. The language throughout has been revised slightly for succinctness, clarity and grammar.

  1. Specify the interviewees' relationship to the interviewer in more detail (e.g., researcher's initials, job title).

The interviewer’s initials are listed in the first sentence of the procedure including basic details on the relationship between the interviewer and participants, further details are described in the supplementary material 3. Some additional information has been included in the manuscript (procedure):

N.A., PhD student and clinical trial coordinator, conducted an explanatory statement over the phone with participants to develop rapport and discuss queries relating to the interview process.

  1. The manuscript's attempt to provide a conceptual framework and cognitive model is commendable. However, there are instances where sentences are complex and may benefit from simplification. For instance, consider rephrasing "individuals developed a changed perspective and approach to life and thus became more outwardly focused" to "individuals gained a new perspective on life and became more outwardly focused."

Thank you for this recommendation, the sentence has been changed as suggested. Also, the discussion has been reviewed and other changes have been made for simplification of sentences.

Overall, the text is well-structured and presents important concepts related to DBS and its impacts on patients with OCD. The grammar and writing style are generally good, but some sentences could be made clearer by breaking them down into shorter sentences. 

The authors would like to thank the reviewer for these important comments and their supportive stance.  

Reviewer 2 Report

Comments for authors:

I reviewed the submission entitled "Phenomenological changes associated with deep brain stimulation for obsessive compulsive disorder: a cognitive appraisal model of recovery.” This manuscript provides a qualitative report of the breadth of DBS-induced change to an individual’s life. Overall, my impression of this manuscript was that it provides an novel contribution to the field of research surrounding OCD and deep brain stimulation. I appreciate the authors’ efforts to thoroughly examine relevant factors that impact an individual’s experience receiving DBS treatment (i.e., appreciation for life, identity, etc.). Also, the clarification that changes are defined relative to an individual’s self-concepts and not their personality. However, I have concerns about the manuscript as submitted, with the most notable surrounding the research question the authors were studying, the model provided in the discussion, and the scholarship in the introduction. I provide specific comments below, which I hope will be useful to the authors and to the action editor.

Introduction

Please consider restructuring the introductory section to improve the manuscript’s overall clarity in paragraph 1 (i.e., “gamut of psycho-social impacts”) and paragraph 5 (“Previous qualitative investigations”).

In the discussion on the cognitive and behavioural mechanisms of OCD, few citations are provided supporting the authors’ perspective on these mechanisms. For example, some key citations here would include Craske et al 2008, 2014, 2022 or Edna Foa’s earlier work. If the authors are only approaching this from a cognitive lens, they should state that outright.

The authors argue in the introduction that the YBOCS is used as a standard definition of response and that measuring response with a single rating scale is inadequate. However, the YBOCS is not the only measure of response used in studies on DBS. For instance, Greenberg et al (2006) used more than just the YBOCS as a measure of improvement, and a meta-analysis (Gadot, 2022) also looked at secondary measures.

In the section on ethical considerations, I think the authors should be more nuanced and structured in their argument. For instance, “Personality changes following DBS are common across movement and psychiatric conditions [7,9].” But in the previous sentence the authors state they lack empirical evidence but they’re common. These statements can coexist with the nuanced point that Thomson et al interviewed clinicians and found evidence for personality changes, but those were short-lived and transient.

The authors state: “DBS targets cognitive-affective-reward networks. Through suppression of pathological thought processes, individuals with OCD gain greater mental clarity to express themselves and behave in a manner that is more akin to their particular desires and needs [7].” The authors then say, “the main intended outcome of DBS for psychiatric conditions is the modification of unwanted pathological thoughts and behaviours [8]” without discussing this mechanism. At the same time mention “use of the term ‘personality’ in this context and thus changes to one’s personality is causing a biased and inaccurate interpretation of changes to the self” after acknowledging above that personality changes are common.

Materials & Methods

Please edit the spacing under the analysis section (i.e., “…definitions of each. For the purpose of the results”).

How were patients selected? Were only responders included?

The research question needs much more explanation. What was actually studied?

In the second paragraph under the procedure they describe their research question: “Here, the research question and themes evolved across the datasets, to enable an implicit and deeper level of interpretation.” How did it evolve across the datasets? By what process?

The authors make a strong statement that a research question or hypothesis would introduce “distortion” or “bias.” However, having a research question is common in qualitative analysis (Fossey, 2002).

In addition to adding clarity regarding what the authors actually studied, I also recommend that they clarify what they mean by “pre-existing framework” when they describe use of the inductive analytical strategy.

Results

The table of themes is excellent. I would like the authors to be clear on how they mapped out and named these themes.

Discussion

The discussion felt quite long to me, and I think it can be pared down. In its place, I would like to see a section on limitations.

The model developed by the researchers seemed to imply causation not warranted by the authors’ methods. For example, “individuals had an initial shift in the feeling of lightness, improved mental clarity and drastic reduction in the intensity of the OCD, all which occurred automatically from the DBS and contributed to a calmer and uplifted state of being.” Could the patients selected have experienced any placebo effects? Could there have been any changes in OCD symptomatology due to the passage of time? Is the model reasonable based on a select group of patients? The methods in the manuscript do not rule these factors out.

In discussing cognitive constructs and vulnerabilities (e.g., Figure 3): Do the authors see any relationship to the learning their participants acquired to the learning proposed to occur in the inhibitory learning theory (e.g., Craske et al., 2009)? (I ask because the authors discuss cognitive and behavioral mechanisms in the introduction.)

The authors state that “We avoided use of the term personality within our interviews, and asked the patients if they had experienced changes to who they are as a person.” I thought this was a great question. I recommend including this in the methods (If I missed this, I apologize).

Please review the manuscript for minor errors and opportunities to improve clarity through grammar and sentence structure. A strong recommendation is to edit the use of semi-colons and commas in favor of breaking longer sentences into two, separate sentences.

Author Response

Reviewer 2

I reviewed the submission entitled "Phenomenological changes associated with deep brain stimulation for obsessive compulsive disorder: a cognitive appraisal model of recovery.” This manuscript provides a qualitative report of the breadth of DBS-induced change to an individual’s life. Overall, my impression of this manuscript was that it provides an novel contribution to the field of research surrounding OCD and deep brain stimulation. I appreciate the authors’ efforts to thoroughly examine relevant factors that impact an individual’s experience receiving DBS treatment (i.e., appreciation for life, identity, etc.). Also, the clarification that changes are defined relative to an individual’s self-concepts and not their personality. However, I have concerns about the manuscript as submitted, with the most notable surrounding the research question the authors were studying, the model provided in the discussion, and the scholarship in the introduction. I provide specific comments below, which I hope will be useful to the authors and to the action editor.

The authors would like to thank the reviewer for their comprehensive review and valuable recommendations. Considering the lack of qualitative investigations within this complex cohort of patients, and the implementation of various qualitative analysis technique, it was deemed suitable to adopt an explorative question to minimise bias and allow the true experiences and perspectives of the participants to be elucidated. The requested changes have been implemented and the manuscript has benefited greatly.

Introduction

Please consider restructuring the introductory section to improve the manuscript’s overall clarity in paragraph 1 (i.e., “gamut of psycho-social impacts”) and paragraph 5 (“Previous qualitative investigations”).

Thank you, this has been changed.

In the discussion on the cognitive and behavioural mechanisms of OCD, few citations are provided supporting the authors’ perspective on these mechanisms. For example, some key citations here would include Craske et al 2008, 2014, 2022 or Edna Foa’s earlier work. If the authors are only approaching this from a cognitive lens, they should state that outright.

The authors have provided a brief description on the phenomenology of OCD symptoms relevant in the appraisal of intrusions and manifestation of intrusions into compulsions. Indeed, the focus is on cognitive models of OCD psychopathology as the introduction reads “from a cognitive perspective, it is contended that dysfunctional beliefs and maladaptive appraisal of intrusions drive the progression and maintenance of symptoms”. Thus, the introduction provides a summary of cognitive processes that influence OCD behaviours, which is expanded in the discussion. It is not within the scope of the report to discuss the breadth of models/ references relating to OCD behaviours.

The authors argue in the introduction that the YBOCS is used as a standard definition of response and that measuring response with a single rating scale is inadequate. However, the YBOCS is not the only measure of response used in studies on DBS. For instance, Greenberg et al (2006) used more than just the YBOCS as a measure of improvement, and a meta-analysis (Gadot, 2022) also looked at secondary measures.

Indeed, clinical trials are now incorporating other clinical and global measures of efficacy, this sentence refers to how clinical response is defined within clinical trials. The definition of clinical response is defined as a 35% improvement in the YBOCS. The authors are suggesting that this criterion may need to be revised to more accurately measure DBS effects.

In the section on ethical considerations, I think the authors should be more nuanced and structured in their argument. For instance, “Personality changes following DBS are common across movement and psychiatric conditions [7,9].” But in the previous sentence the authors state they lack empirical evidence but they’re common. These statements can coexist with the nuanced point that Thomson et al interviewed clinicians and found evidence for personality changes, but those were short-lived and transient.

The word common has been removed to avoid confusion, the statement has been changed to read: personality changes following DBS have been reported across movement and psychiatric conditions [7,10], yet it is necessary to interpret such alterations in keeping with the underlying pathological state.

The statement on personality changes is incorporated to allow a subsequent differentiation between changes across movement disorders and psychiatric disorders.

The authors state: “DBS targets cognitive-affective-reward networks. Through suppression of pathological thought processes, individuals with OCD gain greater mental clarity to express themselves and behave in a manner that is more akin to their particular desires and needs [7].” The authors then say, “the main intended outcome of DBS for psychiatric conditions is the modification of unwanted pathological thoughts and behaviours [8]” without discussing this mechanism. At the same time mention “use of the term ‘personality’ in this context and thus changes to one’s personality is causing a biased and inaccurate interpretation of changes to the self” after acknowledging above that personality changes are common.

The sentence has been changed to remove the word common. This paragraph is intended to discuss that evidence from psychiatric conditions is scarce and cannot be extrapolated from movement disorders as the effects and goals of DBS varies between these groups of patients. Also, the last paragraph of the introduction provides further information on differentiating the effects of DBS on personality/ the person between different cohorts.

Materials & Methods

Please edit the spacing under the analysis section (i.e., “…definitions of each. For the purpose of the results”).

Thank you, this change has been implemented.

How were patients selected? Were only responders included?

The methods explains that participants enrolled in a clinical trial who had reached response were invited to participate, as stated:

Individuals participating in our clinical trial investigating DBS for OCD (ACTRN12612001142820) who had reached clinical response were invited to participate in this additional qualitative follow-up study; their closest family member who had been involved in their care prior to and following DBS therapy were also approached. Six people with OCD classified as responders to DBS and six close family members participated in the study.

The research question needs much more explanation. What was actually studied?

The following research question has been added to the procedure:

Thus, the study investigated the research question; what are the phenomenological changes experienced by people with OCD following response from DBS in relation the self, external world, inter-personal world and attitudes towards well-being?

In the second paragraph under the procedure they describe their research question: “Here, the research question and themes evolved across the datasets, to enable an implicit and deeper level of interpretation.” How did it evolve across the datasets? By what process?

It is understood that this statement may cause confusion, thus has been removed and now reads:

An implicit and latent approach was employed to allow a deeper level of interpretation and minimise distortion or bias.

The authors make a strong statement that a research question or hypothesis would introduce “distortion” or “bias.” However, having a research question is common in qualitative analysis (Fossey, 2002).

This was not our intention to suggest that a research question would introduce bias, rather we have employed qualitative analysis methods to minimise bias. Nevertheless, this statement has been removed.

In addition to adding clarity regarding what the authors actually studied, I also recommend that they clarify what they mean by “pre-existing framework” when they describe use of the inductive analytical strategy.

A pre-existing framework has been defined as i.e., strict hypothesis or narrow lens.

Results

The table of themes is excellent. I would like the authors to be clear on how they mapped out and named these themes.

Thank you for your comment, common themes that applied to all transcripts were identified as described in the analysis, a supplementary figure has been added to demonstrate the step-by-step approach. In sum, the codes from each transcript were reviewed, the most prominent codes then formed part of the thematic map, the thematic map was developed to represent all transcripts. 

Discussion

The discussion felt quite long to me, and I think it can be pared down. In its place, I would like to see a section on limitations.

Apologies for not including this in the original manuscript, the following section of limitations has been added:

There are several limitations within the study. Although we aimed to include the perspectives of all patients within the clinical trial, one was experiencing hypomanic symptoms and one had withdrawn, thus inclusion was deemed inappropriate. The study involved a small sample of 12 participants and should be extended across larger cohorts, and to potentially validate the frameworks/models proposed. Nevertheless, considering the rarity of OCD DBS patients, and lack of phenomenological investigations across DBS applications, we provide a substantial level of scientific evidence to elucidate the patient perspective. Owing to the scarcity of lived experience evidence in the field we adopted a broad research question relating to global functional changes. Following our proposed theories on the cognitive appraisal of intrusions, progression of DBS induced changes, and re-emergence of the self, future investigations should adopt a more informed and specific approach exploring these elements.

The model developed by the researchers seemed to imply causation not warranted by the authors’ methods. For example, “individuals had an initial shift in the feeling of lightness, improved mental clarity and drastic reduction in the intensity of the OCD, all which occurred automatically from the DBS and contributed to a calmer and uplifted state of being.”

Thank you for raising this, our intention is not to imply causation, rather propose theoretical frameworks which describe the phenomenological changes experienced. The discussion has been reviewed and adjusted considering this comment. This statement is now commenced with ‘It is proposed that…”.  Further, sentences have been revised to represent a statement/argument/contention rather than imply causation.

Could the patients selected have experienced any placebo effects?

At baseline, these patients were extremely severe, treatment refractory and debilitated. For example, experiencing symptoms throughout every waking hour, and described their life as a living hell. DBS was offered as a last report. Following DBS, they have achieved dramatic symptomatic, cognitive, and functional changes. It is deemed that placebo effects cannot account for these improvements. Also, the concept of placebo is relevant to the clinical outcomes, which are reported elsewhere.

Could there have been any changes in OCD symptomatology due to the passage of time?

Fluctuations in symptoms are common across OCD DBS patients, as shown in our clinical trial report and from other sites. The focus of this report is not to propose the underlying neurobiological mechanisms of changes in OCD symptomology, rather the focus is on the patient perspective and biopsychosocial factors and processes that contributed to functional recovery. Within this stance, there are various factors that have influenced changes in symptoms, however it is well established that DBS and adjunct psychotherapy cause a dramatic shift in the person, as these individuals were otherwise treatment refractory, chronic, and severely debilitated.

Is the model reasonable based on a select group of patients? The methods in the manuscript do not rule these factors out.

It is not within the scope of the report to comment on all possible contributing and confounding factors, the clinical trial report has addressed controlled and confounding factors that influences symptom changes. Further, this is a lived experience investigation, with the aim to present biopsychosocial factors that contributed to recovery from OCD. A previous investigation by a highly experienced DBS site of OCD (De Haan et al., 2013, 2015) conducted qualitative interviews in 14 patients, and proposed an enactive-affordance model describing the progression of changes experienced. Although our sample is smaller, we employed a more rigorous approach by allowing the qualitative themes to not conform to the interview questions, reached consistency in themes across transcripts (i.e., saturation) and triangulation from a carers perspective. Therefore, it is deemed appropriate to propose the frameworks/models discussed. The authors are conducting an additional lived experience investigation in OCD DBS patients at another site with a pre and post DBS comparison to potentially validate the frameworks/models. Therefore, we have added a suggestion in the limitations describing the need to extend the investigations across a larger sample to validate the findings.

In discussing cognitive constructs and vulnerabilities (e.g., Figure 3): Do the authors see any relationship to the learning their participants acquired to the learning proposed to occur in the inhibitory learning theory (e.g., Craske et al., 2009)? (I ask because the authors discuss cognitive and behavioral mechanisms in the introduction.)

Owing to the cross-sectional and retrospective nature of the interviews, it was not possible to obtain reliable information relating to this theory.

The authors state that “We avoided use of the term personality within our interviews, and asked the patients if they had experienced changes to who they are as a person.” I thought this was a great question. I recommend including this in the methods (If I missed this, I apologize).

Thank you for highlighting this, the following statement has been added to the methods:

To adopt a global lens and avoid bias, the interview did not include terms such as personality and symptoms and rather focused on changes to the person and functioning.

Comments on the Quality of English Language

Please review the manuscript for minor errors and opportunities to improve clarity through grammar and sentence structure. A strong recommendation is to edit the use of semi-colons and commas in favor of breaking longer sentences into two, separate sentences.

Thank you for this recommendation, the language has been revised as suggested for clarity and grammar.  

Round 2

Reviewer 2 Report

I greatly appreciate the work the authors have done on this revision. I have some points that I think can be easily addressed.

1. The authors have a heading titled “The cognitive and behavioural mechanisms of OCD psychopathology are complex.” During the first round of review, I indicated that the authors described the cognitive mechanisms well but not the behavioral mechanisms. This was driven by missing citations from the most popular theories of cognitive and behavioral mechanisms underlying OCD (e.g., inhibitory learning, like Craske et al 2008, 2014, 2022, or Edna Foa’s emotional processing theory).

I do not see how it is outside the scope of a discussion on “cognitive and behavioural mechanisms of OCD psychopathology” to acknowledge this work. Thus, if the authors are only interested in cognitive models of OCD psychopathology, then I would retitle the heading to reflect this and avoid discussion of behavioral elements all together.

2. I agree that a YBOCS cut-off is often (but not always) used to measure clinical response. My point was that since much of the authors’ argument in the paragraph on DBS efficacy hinges on the claim that the “the definition of clinical response is defined as a 35% improvement in the YBOCS,” I suggest they support this. I think a citation is enough to demonstrate the consensus in the field (e.g: Mataix-Cols et al., 2016; https://doi.org/10.1002/wps.20299).

3. “in keeping with the underlying pathological state” is a bit vague. What’s an underlying pathological state? Otherwise, the neuroethical concerns section reads nicely.

4. Thank you for patiently clarifying the participant selection procedure. This was already nicely explained in the first version, as the authors pointed out.

5. The materials and methods sections are much clearer.

6. The supplementary figure is fantastic.

7. The limitations section is good and much appreciated. As an aside, I think the authors overestimate the strength of the study design, but I see no need to add to the limitations to recognize these points. For example, I think the authors and I agree that DBS exerts an effect above and beyond placebo. However, the authors might consider that placebo effects would still be expected during DBS. Both placebo and DBS should jointly lead to positive treatment outcome (e.g., Margo, 1999; https://www.sciencedirect.com/science/article/abs/pii/S0039625799000600). The relevant question is to what extent DBS has over and above placebo.

Author Response

I greatly appreciate the work the authors have done on this revision. I have some points that I think can be easily addressed.

The authors appreciate the thorough review, which has improved the manuscript, thank you again for the valuable feedback.

  1. The authors have a heading titled “The cognitive and behavioural mechanisms of OCD psychopathology are complex.” During the first round of review, I indicated that the authors described the cognitive mechanisms well but not the behavioral mechanisms. This was driven by missing citations from the most popular theories of cognitive and behavioral mechanisms underlying OCD (e.g., inhibitory learning, like Craske et al 2008, 2014, 2022, or Edna Foa’s emotional processing theory).

I do not see how it is outside the scope of a discussion on “cognitive and behavioural mechanisms of OCD psychopathology” to acknowledge this work. Thus, if the authors are only interested in cognitive models of OCD psychopathology, then I would retitle the heading to reflect this and avoid discussion of behavioral elements all together.

The title has been adjusted to include cognitive mechanisms only.

  1. I agree that a YBOCS cut-off is often (but not always) used to measure clinical response. My point was that since much of the authors’ argument in the paragraph on DBS efficacy hinges on the claim that the “the definition of clinical response is defined as a 35% improvement in the YBOCS,” I suggest they support this. I think a citation is enough to demonstrate the consensus in the field (e.g: Mataix-Cols et al., 2016; https://doi.org/10.1002/wps.20299).

Thank you for identifying this, the citation has been added.

  1. “in keeping with the underlying pathological state” is a bit vague. What’s an underlying pathological state? Otherwise, the neuroethical concerns section reads nicely.

Thank you for highlighting this, the sentence has been changed to the following:

Personality changes following DBS have been reported across movement and psychiatric conditions [7,11], yet it is necessary to interpret such alterations in keeping with the pathological mechanisms of that patient group. For example, a personality change in an individual with a movement disorder versus someone with an addiction disorder should be interpreted differently.

The aim of this sentence is to contend that changes in the self should be interpreted in relation to pathology of that patient group. For example a personality change in movement disorders would be a concern, yet a personality change in a psychiatric condition may indeed be beneficial and desirable to the individual (reflecting functioning closer to the individuals true self and further away from pathological constructs).

  1. Thank you for patiently clarifying the participant selection procedure. This was already nicely explained in the first version, as the authors pointed out.

Great, thank you.

  1. The materials and methods sections are much clearer.

Great, thank you.

  1. The supplementary figure is fantastic.

Great, thank you.

  1. The limitations section is good and much appreciated. As an aside, I think the authors overestimate the strength of the study design, but I see no need to add to the limitations to recognize these points. For example, I think the authors and I agree that DBS exerts an effect above and beyond placebo. However, the authors might consider that placebo effects would still be expected during DBS. Both placebo and DBS should jointly lead to positive treatment outcome (e.g., Margo, 1999; https://www.sciencedirect.com/science/article/abs/pii/S0039625799000600). The relevant question is to what extent DBS has over and above placebo.

Thank you for raising this point, the following has been added to the discussion:

It is not within the scope of the report to discuss possible placebo effects. Yet, considering the baseline severity and chronicity of these individuals, and longevity of the treatment effect (up to 9 years), we contend residual placebo effects. Further, it is proposed that the phenomenological changes experienced were an augmented effect of the DBS and adjunct clinical care, and ongoing implementation of cognitive strategies by the participants.
